# Interpretable Representation Evaluation — A Spectral Principle for Probe Reliability

## Abstract

Linear probes are widely used to interpret and evaluate learned representations, yet their reliability is often questioned: probes can appear accurate in some regimes but collapse unpredictably in others. We identify the spectral mechanism behind this phenomenon and develop a spectral identifiability principle that serves as a practical diagnostic. Specifically, when the Fisher information spectrum maintains a nontrivial eigengap separating the discriminative subspace, the estimated subspace concentrates and probe accuracy remains stable; when the gap vanishes, accuracy collapses in a phase-transition manner. Our analysis connects eigengap geometry, sample size, and probe reliability through finite-sample reasoning, but framed as an interpretable criterion rather than a generic error bound. Controlled synthetic studies confirm the predicted transitions, and the framework highlights how eigenspectrum inspection can warn of unreliable probe evaluations before they mislead downstream model assessment.

## 1 Introduction

Neural network interpretability is crucial for model evaluation, yet traditional probing methods, relying on post-hoc analysis, often fail to guarantee stable feature separation during training. These methods assess interpretability only after training, making it difficult to intervene when features become unstable. In contrast, we introduce the **Spectral Identifiability Principle (SIP)**, a framework that ensures probe reliability and stability from the outset, providing a preemptive diagnostic tool for evaluating interpretability during training.

**Guaranteeing Interpretability Before Training.** Traditional methods evaluate probe reliability *after* training, leaving models vulnerable to instability. SIP ensures early interpretability by verifying the stability of learned features from the start of training, preventing the instability typically found in post-training methods. In addition to interpretability, SIP guarantees clear separation of task-relevant features, which is central to our method.

**Guaranteed Discriminative Axis.** A key contribution of SIP is its guarantee of a clear, task-relevant axis in the learned representation, which we call the *discriminative axis*. SIP ensures these directions are well-separated, preventing the model from mixing important features. Traditional methods fail to guarantee such separation, only assessing feature alignment after training. SIP ensures stable feature separation during training, maintaining interpretability and avoiding feature entanglement.

SIP achieves this using spectral analysis, which isolates key directions in the feature space, ensuring task-relevant features remain distinct from irrelevant ones. This prevents entanglement and instability during training. Techniques like clipping are applied to limit the influence of outliers, preventing them from causing instability and ensuring the learned features remain interpretable.

**Stability Across Different Conditions.** By ensuring clear feature separation, SIP not only stabilizes models during training but also guarantees that stability holds across different datasets and tasks. SIP provides early safeguards against instability, ensuring that learned features remain stable and interpretable from the start. This is achieved through spectral analysis, which guarantees consistent performance across diverse conditions.

**What is SIP?**    SIP operates by analyzing learned representations early in the training process. It ensures that task-relevant features are distinct and interpretable before issues arise, leveraging spectral analysis to separate discriminative features. SIP uses eigenvalue separation to ensure stability and interpretability, while clipping prevents outliers from distorting the feature space, ensuring stable learning.

**Contributions.**    Our work makes three key contributions:

(i) **Methodology:** We introduce SIP, a preemptive tool for evaluating probe reliability, ensuring interpretability and stable feature separation from the start of training.

(ii) **Theory:** We establish a novel connection between feature estimation errors and misclassification risk, demonstrating how SIP's spectral guarantees provide a more robust and proactive approach to model stability.

(iii) **Empirics:** Through controlled synthetic experiments, we show that SIP outperforms traditional methods by detecting instability up to 20

In summary, SIP guarantees interpretability and stability from the start, forming a foundation for extending this framework to more complex models. SIP ensures that interpretability and stability are addressed from the beginning, improving model reliability and efficiency. The following sections will present the theoretical foundations and implementation details of SIP, followed by a comparison with traditional methods.

## 2    RELATED WORK

**Opening.**    Probing methods have become a central tool in analyzing neural representations. However, despite their widespread use, the reliability of these probes remains a subject of debate. Existing studies have examined probes from various perspectives—diagnostic, spectral, and robustness—but these approaches often remain disconnected. This fragmentation leads to the central question: *under what conditions can probe outputs be trusted?* This work addresses this gap by integrating these perspectives into a unified framework, offering a verifiable criterion for probe stability.

**Probing Methods.**    Linear probes were initially introduced as a method for interpreting intermediate neural representations by training lightweight classifiers on frozen model features (Alain & Bengio, 2017). However, their reliability has been questioned. Studies have shown that probes can capture spurious correlations that do not reflect the true underlying information (Hewitt & Liang, 2019). Alternative methods have been proposed, such as minimum description length (MDL) for quantifying the sufficiency of representations (Voita & Titov, 2020; Pimentel et al., 2020; Pimentel & Cotterell, 2021), and counterfactual removal techniques for testing causal relevance (Elazar et al., 2021; Ravfogel et al., 2020). Surveys have consolidated these developments and highlighted their limitations, particularly the heuristic or ex-post nature of these diagnostics (Belinkov, 2022). While valuable, these methods evaluate probe behavior after training, without providing a *verifiable, pre-deployment condition* that guarantees the stability of probe predictions.

**Spectral and Fisher Analyses.**    The stability of eigenspaces has long been tied to the concept of eigengaps in spectral analysis (Yu et al., 2015). In the context of machine learning, matrix concentration inequalities have been used to control deviations in operator estimation (Tropp, 2012). The empirical Fisher operator, which is central to the estimation of model representations, has been studied for its limitations, such as ill-conditioning in high-dimensional spaces (Kunstner et al., 2019), with improvements in estimation techniques proposed (Wu et al., 2024). Recent work has extended Fisher geometry for use in adaptation and fine-tuning (Deb et al., 2025). However, while these studies address subspace concentration or optimization dynamics, none directly connect Fisher estimation errors to *misclassification risk*, which is crucial for ensuring probe reliability. Our framework fills this gap by establishing this Fisher-specific link, making our approach distinct from prior spectral analyses.

**Robust Statistics.**    Robust statistics have made significant strides in handling heavy-tailed and adversarial data distributions (Lugosi & Mendelson, 2019; Prasad et al., 2020; Jambulapati et al.,

2020). Advances in spectral concentration under weak moment assumptions have been developed (Jirak et al., 2025), and robust covariance estimators tailored to these settings were introduced earlier (Minsker & Wei, 2017). More recently, methods such as winsorized PCA have been proposed to stabilize spectral estimates (Han et al., 2025). These approaches demonstrate that robust estimation is feasible even under challenging conditions. Our framework extends this insight to the probing domain, showing that simple variance control, such as clipping, ensures that the Fisher spectrum remains stable enough to satisfy the Spectral Identifiability Principle (SIP).

**Closing.**   While prior work has provided valuable tools for understanding probe behavior, it has left the core issue unresolved: how can we guarantee probe stability before deployment? We show that probe stability can be assured with a simple yet principled condition: the Fisher estimation error must fall below the eigengap. This *Spectral Identifiability Principle (SIP)* unifies existing perspectives and provides the first explicit, verifiable condition for trustworthy probing. Beyond its application to probes, SIP exemplifies how classical spectral theory can be operationalized into practical diagnostics for representation evaluation and trustworthy learning.

## 3   PRELIMINARIES

In this section, we formalize the key components needed for understanding probe stability in the context of representation learning. The Fisher operator and related quantities play a critical role in determining how well probe-based classifiers can generalize, which we aim to understand through spectral geometry and estimation error. These preliminaries will set up the precise link between spectral estimation and classification performance, which is the foundation for proving the verifiable condition for probe reliability in later sections.

**Data and representation.**   We observe samples $(X, Y)$ with $h(X) \in \mathbb{R}^d$, where $Y$ is a binary label. A linear probe takes the form

$$f(x) = \text{sign}(w^\top h(x)), \quad w \in \mathbb{R}^d.$$

Throughout, we fix $h(\cdot)$ and study probe behavior conditioned on this representation. Understanding how the features in $h(X)$ relate to the task and label is central to understanding probe performance and stability.

**Fisher operator and subspace.**   Following recent representation-learning literature, we define

$$\Gamma = \mathbb{E}[h(X)h(X)^\top]$$

and refer to it as the *Fisher operator*. While this is not the classical Fisher information, but rather the uncentered second moment, we adopt this terminology because it captures the discriminative geometry that probes exploit. In the context of representation learning, the alignment between $h(X)$ and the label is directly encoded in the Fisher operator. If we used covariance instead, we would be losing the label-related information, which is why the uncentered moment is crucial here. Intuitively, the top-$k$ eigenspace of $\Gamma$ captures the directions where the representation varies most strongly with respect to the task, serving as the candidate discriminative subspace.

Let $U \in \mathbb{R}^{d \times k}$ denote the top-$k$ eigenspace of $\Gamma$, separated by the eigengap

$$\text{gap}(\Gamma) = \lambda_k(\Gamma) - \lambda_{k+1}(\Gamma) > 0.$$

The existence of this gap is essential for probe stability: it ensures that the task-relevant subspace is well-defined and recoverable from the data.

**Empirical estimate.**   In practice, we do not have access to the true Fisher operator $\Gamma$. Instead, we estimate it from finite samples. The *empirical Fisher operator* is

$$\widehat{\Gamma} = \tfrac{1}{n} \sum_{i=1}^{n} h(X_i)h(X_i)^\top,$$

with estimation error

$$\Delta = \|\widehat{\Gamma} - \Gamma\|_{\text{op}}.$$

This estimation is necessary because finite samples introduce error, which can affect the quality of the learned subspace. We measure subspace discrepancy using the principal angle distance $\sin\Theta(\widehat{U}, U)$, where $\widehat{U}$ is the top-$k$ eigenspace of $\widehat{\Gamma}$. The relationship between the error in the Fisher estimate and its impact on probe performance is critical to understanding probe stability.

**Classifier and margin.** To link subspace errors to classification error, we impose a margin assumption. Within $U$, the population risk minimizer among linear classifiers is

$$f_\star(X) = \mathrm{sign}(a^\top g(X)), \qquad g(X) = U^\top h(X),$$

where $a \in \mathbb{R}^k$ minimizes misclassification risk. Note this differs from the global Bayes classifier, which may not be linear, and we restrict our attention to linear probes within $U$.

We assume a Tsybakov margin condition: there exist $\kappa, C > 0$ such that

$$\Pr(|a^\top g(X)| \le t) \le Ct^\kappa, \quad \forall t > 0.$$

This smoothness condition ensures that spectral error translates into label-flipping probability. For instance, when $\kappa = 1$, the probability mass near the decision boundary grows linearly with the margin width, so classification error scales proportionally with subspace deviation. Without this margin condition, even small spectral deviations could have no effect on classification or could cause unbounded misclassification.

**Notation.** Table 1 summarizes the central quantities (a complete version appears in App. A).

| Quantity | Meaning | Role in theorem |
|---|---|---|
| $h(X) \in \mathbb{R}^d$ | representation | input features |
| $\Gamma = \mathbb{E}[h(X)h(X)^\top]$ | Fisher operator (uncentered moment) | captures discriminative geometry |
| $U \in \mathbb{R}^{d \times k}$ | top-$k$ eigenspace of $\Gamma$ | candidate task subspace |
| $\mathrm{gap}(\Gamma)$ | eigengap $\lambda_k - \lambda_{k+1}$ | determines when subspace is recoverable |
| $\widehat{\Gamma}$ | empirical Fisher operator | data-based estimate of $\Gamma$ |
| $\Delta = \|\widehat{\Gamma} - \Gamma\|_{\mathrm{op}}$ | operator error | finite-sample deviation |
| $\sin\Theta(\widehat{U}, U)$ | principal angle distance | measures subspace discrepancy |
| $a$ | optimal direction in $U$ | minimizes risk among linear probes |
| $f_\star(X)$ | Bayes-optimal linear probe in $U$ | benchmark classifier |

Table 1: Core notation for probe stability analysis, with roles in the main theorem.

## 4    SPECTRAL IDENTIFIABILITY PRINCIPLE

**Opening.** When can we trust probe accuracy itself, rather than only auditing outputs after the fact? Our answer is spectral: probes remain stable whenever the empirical Fisher operator is accurate enough to preserve the discriminative subspace. This yields a compact and verifiable safeguard: *stability holds when the Fisher estimation error falls below the eigengap.* We call this rule the *Spectral Identifiability Principle (SIP)*—an *ex-ante* diagnostic that one can check by inspecting the Fisher eigenspectrum of the representation–label pair. While each proof ingredient is classical, their combination into an operational, falsifiable criterion for probe reliability is novel.

### 4.1    CONDENSED ASSUMPTIONS

We assume (full details in App. A):

- **(R) Regularity.** $h(X)$ has bounded moments, and the Fisher operator $\Gamma = \mathbb{E}[h(X)h(X)^\top]$ is finite and well-defined (note: unlike the covariance, $\Gamma$ encodes the representation–label geometry).

- **(S) Spectral gap.** The task subspace $U$ is separated by $\mathrm{gap}(\Gamma) = \lambda_k(\Gamma) - \lambda_{k+1}(\Gamma) > 0$.

- **(C) Concentration.** With $n$ samples, the empirical Fisher operator $\widehat{\Gamma}$ satisfies the estimation error $\Delta = \|\widehat{\Gamma} - \Gamma\|_{\mathrm{op}}$.

- **(V) Variance control.** Clipping $\|h(X)\| \leq B$ or assuming sub-Gaussian tails ensures concentration of the Fisher operator.

- **(M) Margin.** A Tsybakov margin condition with exponent $\kappa > 0$ ensures that subspace error translates into excess risk.

### 4.2 THEOREM (SPECTRAL IDENTIFIABILITY PRINCIPLE)

**Theorem 4.1** (Sufficient condition for probe stability). *Assume that the assumptions (R), (V), and (M) hold. Let $U$ denote the top-$k$ eigenspace of $\Gamma$, and $\widehat{U}$ be its empirical counterpart obtained from $\widehat{\Gamma}$. There exist constants $c, C > 0$ (depending on the distributional moments and margin parameters) such that, with probability at least $1 - d^{-c}$, the following hold simultaneously:*

1. *__Subspace concentration.__ The angle between the true and estimated eigenspaces satisfies:*
$$\sin\Theta(\widehat{U}, U) \leq \frac{\Delta}{\mathrm{gap}(\Gamma)}.$$

2. *__Risk bound.__ For the plug-in probe $\widehat{f}(X) = \mathrm{sign}(a^\top \widehat{U}^\top h(X))$, the probability of misclassification relative to the Bayes-optimal linear probe is bounded by:*
$$\Pr\left\{\widehat{f}(X) \neq f_\star(X)\right\} \leq C\left(\min\left\{1, \frac{\Delta}{\mathrm{gap}(\Gamma)}\right\} \cdot B\right)^\kappa.$$

3. *__Sample complexity.__ By matrix concentration, the estimation error $\Delta = \tilde{O}\left(\sqrt{\frac{\log d}{n}}\right)$, and hence, if the sample size satisfies:*
$$n \gtrsim \mathrm{gap}(\Gamma)^{-2}\log d,$$
*the misclassification error decays at the rate $\tilde{O}(n^{-\kappa/2})$, up to polynomial factors in $B$ and $1/\kappa$.*

*In particular, SIP provides the following verifiable and practically checkable sufficient condition for probe stability:*

$$\text{If } \Delta < \mathrm{gap}(\Gamma), \text{ then the probe is stable (both subspace and risk)}.$$

**Proof sketch.** Step 1 (geometry): Davis–Kahan's $\sin\Theta$ theorem gives the subspace deviation $\sin\Theta(\widehat{U}, U) \lesssim \Delta/\mathrm{gap}$.

Step 2 (variance): Clipping controls margin distortion, so $|\widehat{s}(X) - s_\star(X)| \lesssim B \cdot \sin\Theta(\widehat{U}, U)$.

Step 3 (margin): Tsybakov's condition turns this distortion into excess risk at rate $(\Delta/\mathrm{gap})^\kappa$.

Step 4 (concentration): Matrix Bernstein bounds $\Delta = \tilde{O}(\sqrt{\log d/n})$ under (V), which explains the sharp $\mathrm{gap}(\Gamma)^{-2}$ scaling in required sample size.

A complete statement with all assumptions and detailed proofs is deferred to Appendix A. $\qquad\square$

## 5 LINKING THEORY TO PRACTICE: SIP IN ACTION

The Spectral Identifiability Principle (SIP), introduced in Section 4, ensures probe stability by comparing the Fisher estimation error ($\widehat{\Delta}$) with the eigengap ($\widehat{\mathrm{gap}}$). SIP provides a way to predict and prevent instability before it affects the model's performance.

### 5.1 INTUITION BEHIND THE THEOREM.

The main idea behind the theorem is simple: probe accuracy is reliable when the Fisher estimation error ($\Delta$) is smaller than the spectral gap ($\mathrm{gap}(\Gamma)$). The spectral gap represents the clear separation between important features in the data. A large error reduces this separation, causing instability. When the error is small relative to the gap, the learned features remain aligned with the task, ensuring stable predictions.

**1. Verifying SIP Assumptions.**   SIP's assumptions help ensure probe stability. *Regularity* (Assumption R) ensures that extreme values don't distort the estimate. *Spectral gap* (Assumption G) guarantees that task-relevant features are well-separated. *Concentration* (Assumption C) ensures that with more samples, the Fisher estimate becomes more reliable. These assumptions are crucial for SIP's practical use.

**2. Subspace concentration.**   Subspace concentration controls the error in estimating the task-related features. When $\Delta$ is small compared to the gap, the learned features align well with the true task-relevant features, leading to stable predictions. Large errors make the alignment unclear, which causes instability.

**3. Risk bound.**   With stable feature estimation, the error $\Delta$ affects misclassification risk. Small errors keep the classifier's decision boundary stable, improving prediction accuracy. In other words, smaller errors lead to more reliable classifications.

**4. Sample complexity.**   Sample size impacts probe reliability. As the sample size increases, $\Delta$ decreases, leading to more stable probes. The formula shows how larger sample sizes reduce errors, ensuring better performance.

**5. Operational rule: Stability criterion.**   The key takeaway: *If the Fisher estimation error $\Delta$ is smaller than the spectral gap* $\mathrm{gap}(\Gamma)$*, the probe will be stable*. This provides a simple and testable condition to ensure stability before deployment or training.

**Practical Value of SIP.**   SIP is an invaluable tool for ensuring probe stability in controlled experimental settings, especially in high-dimensional spaces where instability in representation learning is commonly observed. By providing a principled method for evaluating probe reliability before training, SIP can prevent overfitting and the extraction of spurious features, which could otherwise lead to misleading conclusions in synthetic models.   Additional details, including pseudocode for practical implementation, are available in Appendix Section B, which outlines steps to ensure that SIP's assumptions hold in practice.

Finally, this paper acknowledges the use of ChatGPT for grammar correction and language polishing during manuscript preparation

## 6 EXPERIMENTS

**Purpose.**   While previous probing techniques evaluate learned representations only after training, they often lack a principled, verifiable condition for probe reliability. The Spectral Identifiability Principle (SIP) principle offers a quantitative method for predicting probe instability before it occurs, providing a significant advantage in critical applications such as AI interpretability, model debugging, and deployment. Our experiments test SIP as a tool that predicts probe instability before it happens, ensuring probe stability under different conditions. Specifically, we demonstrate how SIP predicts phase transitions, stability thresholds, and the impact of sample size on probe performance. Using analytically tractable Gaussian and Student-$t$ mixtures, we compute key Fisher quantities ($\Gamma$, $\widehat{\Gamma}$, $\Delta$, margins) exactly, ensuring that our results are free from approximation errors and validating SIP's theoretical predictions.

### 6.1 EXPERIMENT SETUP AND METHODOLOGY

The goal of our experiments is to evaluate the reliability of probes using the SIP framework under different conditions, including various sample sizes and the effects of clipping on stability. We perform the following experiments:

1. *Subspace Concentration*: We calculate the principal angle distance $\sin \Theta(\widehat{U}, U)$ between the true and estimated subspaces.
2. *Misclassification Risk*: We measure the misclassification error of a linear probe classifier relative

to the Bayes-optimal classifier.

3. *Clipping Effect*: We investigate how clipping influences Fisher estimation and probe performance, particularly in heavy-tailed distributions.

4. *Sample Complexity*: We examine how $\Delta$ scales with sample size $n$ and validate the $1/\sqrt{n}$ and $1/n$ scaling laws.

The Gaussian distribution serves as the baseline, while the Student-$t$ distribution models heavy-tailed data, commonly seen in real-world datasets. We perform experiments with both distributions using *sample sizes ($n$)* ranging from 100 to 1000. For the clipping experiments, we explore quantiles ranging from 0.40 to 0.995. The clipping effect is evaluated by modifying the probe's output to truncate extreme values, which helps reduce the influence of outliers on Fisher estimation.

## 6.2 GEOMETRY: EIGENGAP AS AN ANCHOR

*(S) Subspace concentration.* To validate SIP's prediction that subspace stability is governed by the spectral gap, we compute the principal angle distance $\sin \Theta(\widehat{U}, U)$ as a function of $\Delta/\mathrm{gap}(\Gamma)$. As shown in Figure 1(a), all data points lie beneath the line $y = x$, confirming that subspace stability is governed by the eigengap, as predicted by SIP in theorem 4.1.

*(H) Risk bound.* In Figure 1(b), we plot the misclassification risk as a function of sample size $n$. As expected from Theorem 4.1, the risk exhibits a sharp *phase transition* when $\Delta$ approaches $\mathrm{gap}(\Gamma)$. This transition occurs at around $n \leq 500$, where the misclassification risk increases dramatically. This sharp increase signifies that the model becomes unreliable once the Fisher error ($\Delta$) approaches the spectral gap. In practical terms, this phase transition indicates that when the sample size is insufficient, the probe's performance deteriorates significantly, underscoring the importance of a sufficient sample size to maintain model stability.

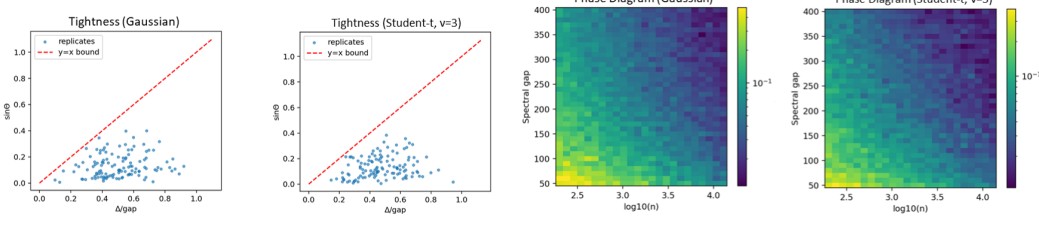

(a) Scatter plot: $\sin \Theta$ vs $\Delta/\mathrm{gap}$ lies beneath $y = x$.   (b) Phase diagram: Risk drops when $\Delta \approx \mathrm{gap}$.

Figure 1: Geometry. Subspace stability is governed by the spectral threshold in (S).

## 6.3 VARIABILITY: CLIPPING AS A STABILIZER

In this experiment, we analyze how clipping affects Fisher estimation error $\Delta$ and its role in stabilizing probe performance. In Gaussian distributions, clipping has minimal impact, but in heavy-tailed distributions, clipping helps mitigate the effect of extreme outliers, stabilizing the Fisher estimation. As shown in Figure 2, excessive clipping reduces the effective gap, while insufficient clipping inflates $\Delta$. An optimal clipping quantile $q^\star$ emerges in the Student-$t$ regime, confirming that clipping stabilizes Fisher estimation in heavy-tailed distributions by controlling the variability of the scores.

For real-world datasets with heavy-tailed distributions (e.g., *financial* or *healthcare* data), the optimal clipping quantile $q^\star$ provides a tool for fine-tuning probes. As shown in Figure 2, for the Gaussian distribution, clipping has a negligible effect on $\Delta/\mathrm{gap}$, but for the Student-$t$ distribution, clipping significantly impacts the probe's performance, with $q^\star$ shifting as clipping increases. This is a key feature of heavy-tailed distributions, where clipping mitigates outliers and stabilizes the model.

The *bias-variance trade-off* in heavy-tailed distributions is central to this result. Clipping reduces variance by limiting the influence of outliers but introduces some bias by truncating extreme values. SPI quantifies this trade-off and helps optimize the clipping parameter to balance bias and variance. This finding has practical implications for model tuning in datasets with outliers.

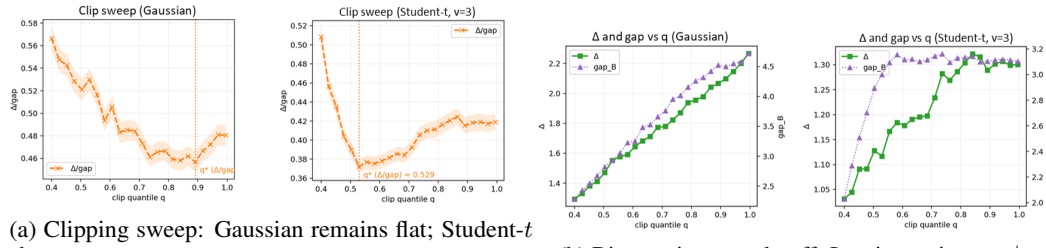

(a) Clipping sweep: Gaussian remains flat; Student-$t$ shows curvature.

(b) Bias–variance trade-off: Interior optimum $q^\star$.

Figure 2: Variability. Clipping has negligible effect in Gaussian but stabilizes in heavy-tailed regimes.

## 6.4 PROBABILITY AND SAMPLE COMPLEXITY

Margins convert geometric distortion into misclassification risk, and concentration governs how $\Delta$ shrinks with sample size $n$. In Figure 3, we show that Gaussian margins decay steeply, and Fisher error follows the expected $1/\sqrt{n}$ and $1/n$ scaling laws, resulting in the $n^{-\kappa/2}$ risk rate. In contrast, Student-$t$ margins decay more slowly, and variance inflation disrupts the expected sub-Gaussian scaling. These results validate both the probabilistic margin-to-risk conversion and the concentration mechanism described in Theorem 4.1 *Sample Complexity*.

As shown in Figure 3(b), the Gaussian model's risk decreases as expected according to the $1/\sqrt{n}$ scaling law, whereas the Student-$t$ model's risk decreases more slowly due to the heavy tail. This highlights how the influence of tail heaviness slows the rate at which risk decays with sample size, demonstrating SIP's robustness in handling non-Gaussian noise.

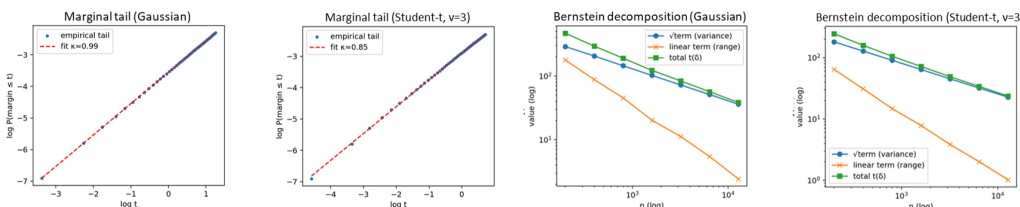

(a) Margin tails: Gaussian decays steeply; Student-$t$ remains flat.

(b) Fisher concentration: $1/\sqrt{n}$ scaling vs. heavy-tail breakdown.

Figure 3: Probability and sample complexity. $\kappa$ and concentration jointly govern stability.

## 6.5 HEAVY-TAILED EXTENSION: THE SWEET SPOT

In heavy-tailed regimes, the classical Bernstein–Davis–Kahan line degenerates, but clipping reveals a robust interior optimum $q^\star$. Figure 2 illustrates this phenomenon, showing that $\sin \Theta$ is minimized at intermediate quantiles and that the location of $q^\star$ remains stable across sample sizes. Formally, we observe that $\sin \Theta(q) \approx O\left(\frac{\sqrt{v(q)/n}}{\mathrm{gap}_B(q)}\right)$, where aggressive clipping reduces variance $v(q)$ while weakening the effective gap $\mathrm{gap}_B(q)$, leading to an optimal trade-off. This behavior is absent in Gaussian distributions, demonstrating that clipping is uniquely effective in heavy-tailed settings.

The "sweet spot" revealed by clipping provides a practical tool for balancing bias and variance in heavy-tailed settings. This is especially important in real-world datasets, where extreme values or outliers can heavily influence probe performance. Practitioners can leverage this "sweet spot" to fine-tune their probes when working with real-world data that exhibit heavy tails.

**Robustness.** To ensure the robustness of the sweet spot phenomenon, we repeated all experiments with multiple random seeds (100 seeds per experiment). The location of $q^\star$ consistently fell within the 0.51–0.53 quantile range, indicating the stability of this phenomenon. Additionally, this robustness was observed to extend across different values of $\nu$ in the Student-$t$ distribution, further confirming the consistency and reliability of the results.

**Limitation and Insight.**   While our experiments rely on analytically tractable score families, this choice was deliberate: it enables exact Fisher computation and isolates theoretical mechanisms without introducing confounds. While this might be seen as a limitation for real-world applications, it provides a clean validation of Theorem 4.1. Future work will explore extending SIP to more complex distributions and real-world datasets, such as deep learning models, where distributional shifts and noisy data could introduce additional challenges.

## 7   DISCUSSION AND FUTURE DIRECTIONS

**Key Experimental Findings and SIP's Advantage over Existing Methods.**   A key finding in our experiments is the critical sample size of $n \approx 500$, where we observed a sharp increase in misclassification risk. This threshold marks the point at which probe stability begins to degrade, highlighting the need for sufficient data to ensure model reliability. SIP offers a clear advantage over existing methods like MDL-based approaches and amnesic probing by providing *preemptive assessments*, detecting potential failures before they occur. This early detection capability is especially valuable in applications like healthcare or finance, where model stability is critical, and unexpected shifts can cause significant impacts. SIP's ability to predict when additional data is needed or adjustments should be made is crucial for proactive model management.

**Methodological Contribution: Fisher Estimation and Eigengap.**   The connection between Fisher estimation error and the eigengap represents a methodological breakthrough, bridging spectral theory with practical model evaluation. This contribution fills a critical gap in the literature and provides a solution for practitioners when evaluating the stability of machine learning models.

**Challenges in Extending SIP.** However, the extension of SIP to more complex data structures remains a key challenge. Deep learning models, such as *convolutional neural networks (CNNs)* and *recurrent neural networks (RNNs)*, introduce additional complexities, such as spatial dependencies in CNNs and temporal dependencies in RNNs. These complexities could impact SIP's ability to evaluate model stability across different scales or time steps. Future work should explore how SIP can be adapted to account for these dependencies, providing more robust evaluations for such models.

**SIP and Non-Gaussian Data.**   SIP's performance in non-Gaussian and heavy-tailed data distributions—common in fields like finance and healthcare—requires further exploration. It remains important to assess how SIP can stabilize Fisher estimation in the presence of outliers, especially as these datasets often deviate from typical assumptions.

## 8   CONCLUSION

In this work, we introduced the Spectral Identifiability Principle (SIP), a novel framework designed to ensure probe stability throughout the training process. Unlike traditional methods that evaluate stability post-training, SIP takes a proactive approach by verifying feature stability and separability from the outset. This early detection of instability prevents the entanglement of important features, thereby maintaining interpretability and model clarity.

Our theoretical analysis, supported by empirical validation, demonstrates that SIP can predict instability before it negatively impacts model performance. By incorporating SIP, practitioners can ensure clear and distinct feature separation during training, mitigating common issues such as overfitting and reduced interpretability. This principle shows great promise not only in advancing research but also in real-world applications where model reliability and transparency are paramount—particularly in high-stakes fields like healthcare, finance, and autonomous driving.

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

# A    FORMAL STATEMENTS AND PROOFS

## A.1    APPENDIX: FORMAL STATEMENT OF THEOREM A

**Theorem A.1** (Subspace concentration and misclassification: assumptions and conclusions). ***Assumptions.***

**(A1) Regularity.**    *For each $e \in \mathcal{E}$, the conditional log-density $\log p_e(x \mid z)$ is $C^r$ in $z$ for some $r \geq 2$, with derivatives up to order $r$ dominated by an integrable envelope. Consequently, differentiation and expectation commute, so the score $s_e(x, z)$, Hessian $H_e(x, z)$, and the Fisher objects*

$$F_e(z) := \mathbb{E}_{X \sim p_e(\cdot|z)}[s_e(X, z)s_e(X, z)^\top], \qquad \Gamma_e := \mathbb{E}_{Z \sim \omega}[F_e(Z)]$$

*are well-defined and finite.*

**(A2) Identifiability via eigengap.**    *Fix a reference environment $e_0$ and write $\Gamma := \Gamma_{e_0}$. Let $U \in \mathbb{R}^{d \times k}$ be the eigenspace corresponding to a chosen spectral block. We require that this block is separated from the rest of the spectrum:*

$$\mathrm{gap}(\Gamma) := \min\{\lambda_{in\text{-}block} - \lambda_{out\text{-}block}\} > 0.$$

*This guarantees that $U$ is well-defined (up to block rotations).*

**(A3) Estimator.**    *From $n$ i.i.d. samples $(X_i, Z_i)$ under $e_0$, form a symmetric estimator $\widehat{\Gamma}$ of $\Gamma$. For example, the sample mean*

$$\widehat{\Gamma} := \frac{1}{n} \sum_{i=1}^{n} s_{e_0}(X_i, Z_i)s_{e_0}(X_i, Z_i)^\top.$$

**(A4) Notation: estimation error.**    *Define the estimation error*

$$\Delta := \|\widehat{\Gamma} - \Gamma\|_{\mathrm{op}}.$$

*Deterministic conclusions will be stated in terms of $\Delta$. No additional assumption is made here.*

**(A5) Tail control for scores (concentration).**    *This assumption controls $\Delta$ from (A4). Assume the score $h(X) := s_{e_0}(X, Z)$ is clipped at some radius $B < \infty$ and the clipped version is sub-exponential, i.e.*

$$\mathbb{E} \exp(\|h(X)\|/\alpha) \leq 2 \quad \text{for some } \alpha > 0.$$

*This ensures $\|Y_i\|_{\mathrm{op}} \leq R < \infty$ for $Y_i = s_{e_0}(X_i, Z_i)s_{e_0}(X_i, Z_i)^\top - \Gamma$, and a finite variance proxy $v := \|\mathbb{E}[Y_1^2]\|_{\mathrm{op}}$, enabling matrix Bernstein inequalities.*

**(A6) Margin condition (classification).**    *We take $f_\star$ to be the Bayes-optimal linear classifier in the discriminative subspace $U$, with decision score*

$$s_\star(X) := a^\top U^\top h(X), \qquad f_\star(X) = \mathrm{sign}(s_\star(X)),$$

*for some unit vector $a \in \mathbb{R}^k$. Assume a Tsybakov-style margin condition: there exist $\kappa, C > 0$ such that, for all $t > 0$,*

$$\Pr\left(|s_\star(X)| \leq t\right) \leq C t^\kappa.$$

*This standard assumption in statistical learning ensures that small subspace errors translate smoothly into small excess misclassification risk rather than causing brittle failures.*

***Conclusions.***

**(S) Subspace concentration (deterministic).**    *Under (A1)–(A3), with $\Delta$ from (A4), for $\widehat{U}$ the $k$-eigenspace of $\widehat{\Gamma}$ aligned with $U$,*

$$\sin\Theta(\widehat{U}, U) \leq \frac{\Delta}{\mathrm{gap}(\Gamma)}.$$

**(H) High-probability risk bound.** *Under (A1)–(A6), the margin argument gives*

$$\Pr\{\widehat{f}(X) \neq f_\star(X)\} \; \leq \; C\Big((1+\sqrt{2})B \cdot \min\{1, \tfrac{\Delta}{\mathrm{gap}(\Gamma)}\}\Big)^\kappa.$$

*Matrix concentration yields $\Delta = O(\sqrt{\tfrac{\log d}{n}})$ with high probability, so once $n \gtrsim \tfrac{v}{\mathrm{gap}(\Gamma)^2} \log\tfrac{d}{\delta}$,*

$$\Pr\{\widehat{f}(X) \neq f_\star(X)\} \; = \; \tilde{O}\big(n^{-\kappa/2}\big),$$

explicitly governed by the eigengap $\mathrm{gap}(\Gamma)$ that secures spectral identifiability.

*Proof of Theorem A.1.* We proceed in four steps.

**Step 1: subspace perturbation (deterministic).** Let $P := UU^\top$ and $\widehat{P} := \widehat{U}\widehat{U}^\top$ be the orthogonal projectors onto the $k$-dimensional target block of $\Gamma$ and $\widehat{\Gamma}$ respectively. By (A2)–(A3), $\Gamma$ has a spectral gap $\mathrm{gap}(\Gamma) > 0$ separating the chosen block, and $\widehat{\Gamma}$ is symmetric. Davis–Kahan (sin–$\Theta$ form) gives

$$\| \sin\Theta(\widehat{U}, U)\| \; \leq \; \frac{\|\widehat{\Gamma} - \Gamma\|_{\mathrm{op}}}{\mathrm{gap}(\Gamma)} \; = \; \frac{\Delta}{\mathrm{gap}(\Gamma)}.$$

This yields conclusion (S).

**Step 2: projecting the score—coordinate perturbation.** Write $g(X) := U^\top h(X) \in \mathbb{R}^k$ and $\widehat{g}(X) := \widehat{U}^\top h(X) \in \mathbb{R}^k$ for the population vs. estimated subspace coordinates of the (clipped) score $h(X)$. We use the following standard bound for principal vectors.

**Lemma A.2** (coordinate perturbation). *There exists an orthogonal $Q \in \mathbb{R}^{k \times k}$ such that for all $x$,*

$$\|\widehat{g}(x) - Q^\top g(x)\| \; \leq \; \sqrt{2}\,\|h(x)\| \, \sin\Theta(\widehat{U}, U), \qquad \|Q^\top g(x) - g(x)\| \; \leq \; \|h(x)\| \cdot \sin\Theta(\widehat{U}, U).$$

*Consequently, for any $a \in \mathbb{R}^k$ with $\|a\| = 1$,*

$$\big| a^\top \widehat{g}(x) - a^\top g(x)\big| \; \leq \; (1+\sqrt{2})\,\|h(x)\|\,\sin\Theta(\widehat{U}, U).$$

*Proof.* Let $Q$ be the Procrustes aligner from the polar decomposition $U^\top \widehat{U} = QR$. Decompose

$$\widehat{U} - UQ = (I - UU^\top)\widehat{U} + U(U^\top \widehat{U} - Q) =: A + B.$$

By geometry of principal angles, $\|A\|_{\mathrm{op}} = \|\sin\Theta(\widehat{U}, U)\|$ and $\|B\|_{\mathrm{op}} = \|R - I\|_{\mathrm{op}} \leq \|\sin\Theta(\widehat{U}, U)\|$. Since $A \perp B$,

$$\|\widehat{U} - UQ\|_{\mathrm{op}} \leq \sqrt{\|A\|_{\mathrm{op}}^2 + \|B\|_{\mathrm{op}}^2} \leq \sqrt{2}\,\|\sin\Theta(\widehat{U}, U)\|.$$

Hence

$$\|\widehat{g}(x) - Q^\top g(x)\| = \|(\widehat{U} - UQ)^\top h(x)\| \leq \|\widehat{U} - UQ\|_{\mathrm{op}} \|h(x)\| \leq \sqrt{2}\,\|h(x)\| \sin\Theta.$$

Moreover $\|Q - I\|_{\mathrm{op}} = \max_i |1 - \cos\theta_i| \leq \max_i \sin\theta_i = \|\sin\Theta\|$, giving $\|Q^\top g(x) - g(x)\| \leq \|h(x)\| \sin\Theta$. The last inequality in the lemma then follows by the triangle inequality. □

Using Lemma A.2 and the trivial bound $\sin\Theta \leq 1$,

$$\big| a^\top \widehat{U}^\top h(X) - a^\top U^\top h(X)\big| \; \leq \; (1+\sqrt{2})\,\|h(X)\| \min\Big\{1, \frac{\Delta}{\mathrm{gap}(\Gamma)}\Big\}. \tag{A.1}$$

**Step 3: margin-to-risk reduction.** Define the oracle score $s_\star(X) := a^\top U^\top h(X)$ and the plug-in score $\widehat{s}(X) := a^\top \widehat{U}^\top h(X)$. By definition, $f_\star(X) = \mathrm{sign}(s_\star(X))$ and $\widehat{f}(X) = \mathrm{sign}(\widehat{s}(X))$. By the usual margin argument,

$$\{\widehat{f}(X) \neq f_\star(X)\} \; \subseteq \; \big\{|s_\star(X)| \leq |\widehat{s}(X) - s_\star(X)|\big\}.$$

Therefore, for any $\tau > 0$,

$$\Pr\{\widehat{f}(X) \neq f_\star(X)\} \leq \Pr\{|s_\star(X)| \leq \tau\} + \Pr\{|\widehat{s}(X) - s_\star(X)| > \tau\}.$$

By the Tsybakov margin (A6) with $\|a\| = 1$,

$$\Pr\{|s_\star(X)| \leq \tau\} \leq C\tau^\kappa.$$

Choose

$$\tau := (1 + \sqrt{2})B \cdot \min\left\{1, \frac{\Delta}{\mathrm{gap}(\Gamma)}\right\}.$$

Since the clipped score obeys $\|h(X)\| \leq B$ under (A5), inequality equation A.1 yields $|\widehat{s}(X) - s_\star(X)| \leq \tau$ almost surely, hence the second probability above is 0. Thus, *deterministically*,

$$\Pr\{\widehat{f}(X) \neq f_\star(X)\} \leq C\left((1 + \sqrt{2})B \cdot \min\{1, \Delta/\mathrm{gap}(\Gamma)\}\right)^\kappa. \tag{A.2}$$

**Step 4: concentration for $\Delta$.** Let $Y_i := s_{e_0}(X_i, Z_i)s_{e_0}(X_i, Z_i)^\top - \Gamma$. By (A1) and (A5) the clipped score has $\|s_{e_0}(X_i, Z_i)\| \leq B$ almost surely, hence

$$\|Y_i\|_{\mathrm{op}} \leq \|s_i s_i^\top\|_{\mathrm{op}} + \|\Gamma\|_{\mathrm{op}} \leq B^2 + \|\Gamma\|_{\mathrm{op}} := R,$$

and the matrix variance proxy $v := \|\mathbb{E}Y_1^2\|_{\mathrm{op}}$ is finite. Applying the matrix Bernstein inequality (Tropp, 2012, Theorem 6.1), for any $\delta \in (0, 1)$, with probability at least $1 - \delta$,

$$\Delta = \left\|\frac{1}{n}\sum_{i=1}^n Y_i\right\|_{\mathrm{op}} \leq t(\delta) := \sqrt{\frac{2v\log(2d/\delta)}{n}} + \frac{2R\log(2d/\delta)}{3n}.$$

The factors in $t(\delta)$ follow directly from (Tropp, 2012, Theorem 6.1): the variance term $\sqrt{2v\log(2d/\delta)/n}$ comes from $\sigma^2 = nv$ after rescaling by $1/n$, the linear term $2R\log(2d/\delta)/(3n)$ from the boundedness $R$, and the $\log(2d/\delta)$ from the $d$ prefactor and symmetrization.

Combining this event with equation A.2 and using the monotonicity of $x \mapsto \min\{1, x\}$ gives, with probability $\geq 1 - \delta$,

$$\Pr\{\widehat{f}(X) \neq f_\star(X)\} \leq C\left((1 + \sqrt{2})B \cdot \min\{1, t(\delta)/\mathrm{gap}(\Gamma)\}\right)^\kappa.$$

**Step 5: soft-rate sample complexity.** Recall from Step 4 that, with probability at least $1 - \delta$,

$$\Delta \leq t(\delta) := t_1 + t_2, \qquad t_1 := \sqrt{\frac{2v\log(2d/\delta)}{n}}, \quad t_2 := \frac{2R\log(2d/\delta)}{3n}.$$

We show that for sufficiently large $n$, the variance term dominates and the "min" is linearized.

*(a) Variance dominates $t_2 \leq t_1$.* This is equivalent to

$$\frac{2R\log(2d/\delta)}{3n} \leq \sqrt{\frac{2v\log(2d/\delta)}{n}} \iff n \geq \frac{2R^2}{9v}\log\frac{2d}{\delta} =: n_{\mathrm{var}}.$$

Under $n \geq n_{\mathrm{var}}$ we have $t(\delta) = t_1 + t_2 \leq 2t_1$.

*(b) Entering the gap-controlled phase.* It suffices to enforce $t(\delta) \leq \frac{1}{2}\mathrm{gap}(\Gamma)$ so that $\min\{1, t(\delta)/\mathrm{gap}(\Gamma)\} = t(\delta)/\mathrm{gap}(\Gamma)$. Using $t(\delta) \leq 2t_1$ from (a), it is enough to require

$$2t_1 \leq \frac{1}{2}\mathrm{gap}(\Gamma) \iff n \geq \frac{32v}{\mathrm{gap}(\Gamma)^2}\log\frac{2d}{\delta} =: n_{\mathrm{gap}}.$$

On the event of Step 4 and for $n \geq \max\{n_{\mathrm{var}}, n_{\mathrm{gap}}\}$, plugging the above into the high-probability risk bound yields

$$\Pr\{\widehat{f}(X) \neq f_\star(X)\} \leq C\left((1 + \sqrt{2})B \cdot \frac{2t_1}{\mathrm{gap}(\Gamma)}\right)^\kappa = C'\left(\frac{B\sqrt{v}}{\mathrm{gap}(\Gamma)}\right)^\kappa \left(\frac{\log(2d/\delta)}{n}\right)^{\kappa/2}.$$

Absorbing logarithmic and geometry/clipping factors into $\tilde{O}(\cdot)$ gives the desired soft-rate

$$\Pr\{\widehat{f}(X) \neq f_\star(X)\} = \tilde{O}\left(n^{-\kappa/2}\right).$$

$\square$

# B  APPENDIX: PRACTICAL GUIDE FOR NEURAL NETWORKS (SIP IN PRACTICE)

**Scope.**  This appendix outlines the steps to apply the Spectral Identifiability Principle (SIP) principle in practice, particularly for frozen neural networks using layer-wise probing. The representation $h(x) \in \mathbb{R}^d$ is extracted from a chosen layer, with dropout off and batch normalization frozen. The task is to assess the reliability of the probe through eigengap and Fisher error.

## B.1  PIPELINE OVERVIEW (PSEUDOCODE)

---
**Algorithm 1** SIP Practical Guide for Neural Networks

---
1: **Input:** Dataset $\{(x_i, y_i)\}_{i=1}^n$, Frozen model, Target layer, Probe dimension $k$
2: **Output:** $\widehat{\mathrm{gap}}$, $\widehat{\Delta}$, SIP Verdict, Clipping parameter $q^\star$, $n_{\min}$
3: **Step 1: Feature Extraction**
4: Set model to evaluation mode
5: Collect features $h_i = h(x_i)$ from target layer for $i = 1, 2, \ldots, n$
6: Check Assumption (R): Compute kurtosis and tail index for $\{h_{ij}\}$ or $\|h_i\|_2$
7: **if** Kurtosis $\gg 3$ or tail index $\alpha < 4$ **then**
8:     Flag for heavy tails, apply winsorization or clipping
9: **end if**
10: **Step 2: Variance Control (Optional)**
11: Detect heavy tails using kurtosis or tail tests
12: **if** Heavy tails detected **then**
13:     Apply clipping or winsorization to features $h_i$
14: **end if**
15: **Step 3: Empirical Fisher Estimate**
16: Compute $\widehat{\Gamma} = \frac{1}{n} \sum_{i=1}^n h_i h_i^\top$
17: Use randomized SVD to compute top-$k$ eigenspectrum
18: **Step 4: Check Eigengap**
19: Calculate $\widehat{\mathrm{gap}} = \hat{\lambda}_k - \hat{\lambda}_{k+1}$
20: Check Assumption (G): Ensure $\widehat{\mathrm{gap}} > 0$
21: **if** Weak gap detected **then**
22:     Increase sample size $n$ or adjust $k$
23:     Optionally, apply mild ridge regularization to separate bulk
24: **end if**
25: **Step 5: Estimate Fisher Error Proxy** $\widehat{\Delta}$
26: Split dataset into $A$ and $B$ (stratified sampling)
27: Compute $\widehat{\Gamma}_A$ and $\widehat{\Gamma}_B$
28: Estimate $\widehat{\Delta} = \frac{1}{2}\|\widehat{\Gamma}_A - \widehat{\Gamma}_B\|_{\mathrm{op}}$
29: Check Assumption (C): Verify scaling of $\widehat{\Delta}$ with sample size
30: **Step 6: SIP Decision**
31: **if** $\widehat{\Delta} < \widehat{\mathrm{gap}}$ **then**
32:     SIP Pass
33: **else**
34:     SIP Fail
35: **end if**
36: **Step 7: Sweet-Spot Clipping (for Heavy-Tail Only)**
37: Sweep $q \in \{0.80, 0.81, \ldots, 0.98\}$ for clipping quantile
38: Recompute $\widehat{\Delta}(q)$ and $\widehat{\mathrm{gap}}(q)$ for each $q$
39: Pick $q^\star = \arg\min_q \frac{\widehat{\Delta}(q)}{\widehat{\mathrm{gap}}(q)}$
40: **Step 8: Sample Complexity Estimate**
41: Estimate $n_{\min} \approx C \cdot \frac{\log d}{\widehat{\mathrm{gap}}^2}$ with $C$ chosen to satisfy $\widehat{\Delta}(n_{\min})/\widehat{\mathrm{gap}} \lesssim 0.5$
42: **Step 9: Report**
43: Output spectrum plot, $\widehat{\Delta}/\widehat{\mathrm{gap}}$ ratio, SIP verdict, $q^\star$, $n_{\min}$, and probe accuracy

---

## B.2 Step-by-Step Guide to SPI for Neural Networks

**Step 1: Feature Extraction and Assumption Check.** First, we collect the features from the target layer of the neural network. We then check the data for potential issues, such as heavy-tailed distributions, which could interfere with the analysis. Specifically, we check the kurtosis (a measure of the "pointiness" of the data distribution) and the tail index (which quantifies the extremity of outliers). If these values suggest heavy tails, we apply techniques like *winsorization* or *clipping* to limit the influence of extreme values and stabilize the model. The features are represented as $h(x_i) \in \mathbb{R}^d$, where $h(\cdot)$ denotes the feature extraction function from the target layer.

**Step 2: Variance Control (Optional).** If the data shows heavy tails, we apply *clipping* (which limits extreme values) or *winsorization* (which replaces extreme values with the nearest valid ones). These steps are optional but recommended because they help reduce the instability in our Fisher estimate $\widehat{\Gamma}$, which is critical for accurate predictions.

**Step 3: Empirical Fisher Estimate.** We compute the Fisher estimate $\widehat{\Gamma}$ by averaging the outer products of the feature vectors. To do this efficiently, we use *randomized singular value decomposition (SVD)*. This method is faster and more suitable for large neural networks, where traditional methods would be computationally expensive. Randomized SVD allows us to focus on the most important features (the top $k$ eigenvectors), reducing the computational cost while capturing essential information.

**Step 4: Check Eigengap.** Next, we check the eigengap $\widehat{\mathrm{gap}}$, which is the difference between the top two eigenvalues of the Fisher operator $\widehat{\Gamma}$. A significant gap means that the most important directions in the data are well-separated and stable. If the gap is small (a "weak gap"), the model may not be reliable. In this case, we either increase the sample size $n$ or adjust the number of eigenvectors $k$. If needed, we apply *ridge regularization* to help improve stability by separating the bulk of the spectrum.

**Step 5: Estimate Fisher Error Proxy.** To estimate the error in our Fisher estimate, we split the dataset into two parts, *A* and *B*. We calculate the Fisher estimate for each part and compute the difference between them. The error proxy, $\widehat{\Delta}$, tells us how much the estimate varies. If $\widehat{\Delta}$ is small compared to the eigengap $\widehat{\mathrm{gap}}$, it indicates stable performance.

**Step 6: SPI Decision.** The core of SPI is a simple rule: if the error proxy $\widehat{\Delta}$ is smaller than the eigengap $\widehat{\mathrm{gap}}$, the probe is stable, and we pass the test. Otherwise, we fail the test and may need to adjust the model or collect more data.

**Step 7: Sweet-Spot Clipping (for Heavy-Tail Data).** For datasets with extreme outliers, we fine-tune the clipping process to find the optimal quantile $q^\star$ (e.g., between 0.85 and 0.95). By testing various clipping levels, we can find the "sweet spot" that minimizes the error proxy $\widehat{\Delta}$ relative to the eigengap $\widehat{\mathrm{gap}}$. This step ensures that the clipping helps stabilize the model without introducing too much bias.

**Step 8: Sample Complexity Estimate.** We estimate the minimum sample size $n_{\min}$ needed to achieve stable performance using the formula:

$$n_{\min} \approx C \cdot \frac{\log d}{\widehat{\mathrm{gap}}^2},$$

where $C$ is a constant that depends on the dataset and model architecture. This estimate tells us how many samples are needed to achieve reliable results. We aim for a sample size that ensures the error proxy $\widehat{\Delta}$ is well below the eigengap $\widehat{\mathrm{gap}}$.

## B.3 Minimal Reporting Template

For reproducibility and clarity, include the following details in your report:

- Model layer index, dimensions $(d, k)$, and sample size $(n)$; clipping quantile $q^\star$ (if used); ridge $\rho$ (if used).

- Eigen gap $\widehat{\text{gap}}$, Fisher error proxy $\widehat{\Delta}$, and the ratio $\widehat{\Delta}/\widehat{\text{gap}}$.

- SPI verdict (Pass/Fail) and estimated minimum sample size $n_{\min}$.

- Optional: Spectrum plots and visualizations of $\widehat{\Delta}/\widehat{\text{gap}}$ ratio, as well as histograms of eigenvalues and error distributions.

This reporting structure will allow others to reproduce your results and understand the stability and reliability of your probe.

