# OpenReview forum: "Interpretable Representation Evaluation — A Spectral Principle for Probe Reliability"
_ICLR.cc/2026/Conference — Submitted to ICLR 2026_

### Official Review · Reviewer_D12E · 2025-10-23

**Soundness:** 1
**Presentation:** 2
**Contribution:** 1
**Rating:** 0
**Confidence:** 5

**Summary:**

The paper introduces the Spectral Identifiability Principle (SIP), a method meant to both measure and ensure linear probe reliability for neural network models.

The work presents a theoretical result about the relation between estimation error and the gap between the k'th and (k+1)'th eigenvalue.

Experiments are run on Gaussian and student-t distributions.

**Strengths:**

**S1:** It would be extremely useful to be able to measure and ensure reliability of linear probes.

**Weaknesses:**

**W1:** The claims presented in the abstract and the introduction are not supported by the rest of the paper.
For example in the introduction is says that SIP is a "diagnostic tool for evaluating interpretability during training" (line 31-32),
"SIP ensures stable feature separation during training, maintaining interpretability and avoiding feature entanglement" (line 42-43)
and "SIP not only stabilizes models during training but also guarantees that stability holds across different datasets and tasks" (line 49-50).
Similar claims in lines 34-37, 54, 70-72, 294-295 and 469-470.
However, there are no experiments in the paper on trained models.


In line 67-68 it also says:
"Through controlled synthetic experiments, we show that SIP outperforms traditional methods by detecting instability up to 20".
However, there is no comparison with any other methods in the paper.


**W2:** The paper has many logical inconsistencies. For example:
- At the beginning and the end of the paper the focus seems to be interpretability, but in the middle it suddenly shifts to be about performance (line 261-262).
- The paper lists 5 assumptions before the main theorem, but then only assume 3 of them in their theorem statement.
- The main result is different in the main paper line 236 compared to in the appendix line 596.

**Questions:**

Since this paper does not give any support for its main claims I recommend rejection.

**Q1:** In line 67-68 it says: "Through controlled synthetic experiments, we show that SIP outperforms traditional methods by detecting instability up to 20".
Up to 20 what?

**Details Of Ethics Concerns:**

As I wrote in my previous comment I suspect this paper was written almost exclusively using a large language model.

I repeat my reasons below:

There are many logical inconsistencies. For example, it says many places in the paper that their method (SIP) guarantees interpretability before/during/throughout training (lines 034-037, 049-052, 054, 070-072, 294-295, 469-470), but there is no mention of any models actually being trained in the experiments. Another example is that in the beginning and at the end of the paper the focus seems to be interpretability, but in the middle it suddenly shifts to be about performance (line 261-262). A third example is that they list 5 assumptions before their theorem, and then only assume 3 of them in their theorem statement. A fourth example: their main result is different in the main paper line 236 compared to in the appendix line 596.

There are weird formatting choices. For example, they state using ChatGPT in line 300-301at the end of section 5 just before the experiment section. Another example is that in section B.3. of the appendix the text changes to second person. In line 809 it says: "For reproducibility and clarity, include the following details in your report:"

---

### Official Review · Reviewer_uqN9 · 2025-10-28

**Soundness:** 1
**Presentation:** 2
**Contribution:** 3
**Rating:** 2
**Confidence:** 2

**Summary:**

The paper introduces the Spectral Identifiability Principle. It is a framework which proposes to study the empirical fisher operator, which is defined as the uncentered second moment of model activations. The paper gives a theorem, ostensibly proving that a probe will be stable whenever the estimation error (operator norm of the difference between the empirical and true fisher operator) is smaller than the eigengap of the fisher operator. Some experiments are provided to give strength to the required assumptions.

**Strengths:**

The proposed method seems interesting. If it can indeed detect whether meaningful directions in latent space are well-separated, then it seems useful for evaluating the interpretability of models.

**Weaknesses:**

The paper makes frequent reference to the idea of (probe) stability. However, this is not explicitly defined. Perhaps it is a well-known property of eigenspaces as suggested by lines 96-97, but for those not familiar a definition would be welcome. As it stands it is unclear to me exactly what problem the paper is trying to solve.

The introduction (line 36) states 'the instability typically found in post-training methods', but nothing is cited to back up this claim.

Section 6 seems to be missing a lot of the experimental details. What model and what data are these experiments performed on?
From line 432-437, it seems like these experiments are not performed on real model activations, is that correct? If so, what exactly is serving as a substitute for said activations, and under what assumptions would we expect the results to generalize to activations obtained from real models/data?

**Questions:**

On line 146, it is stated that 'if we used covariance instead, we would be losing the label-related information, which is why the uncentered moment is crucial here'. Why is that? what about centering the activations would destroy that information, or am I misunderstanding? Seems to contradict probing methods that do in fact center activations such as Burns et. al. (2023).

Line 343 says 'the risk exhibits a sharp phase transition when Delta approaches gap(Gamma)'. What makes this a phase transition, and how do I tell it is sharp? What would the figure look like if it wasn't sharp or not a phase transition? Is it correct that the y-axis displays $k$?

Line 68 seems to contain a cut-off sentence.

Since the estimation error requires both the estimated and true fisher operator, how can we evaluate it in real-world circumstances?


Burns, C., Ye, H., Klein, D., & Steinhardt, J. (2023, February 1). Discovering Latent Knowledge in Language Models Without Supervision. Proceedings of the Eleventh International Conference on Learning Representations. The Eleventh International Conference on Learning Representations. https://openreview.net/forum?id=ETKGuby0hcs

---

### Official Review · Reviewer_m6cR · 2025-10-30

**Soundness:** 2
**Presentation:** 2
**Contribution:** 1
**Rating:** 2
**Confidence:** 4

**Summary:**

The authors highlight a spectral condition under which linear probes can accurately and robustly recover a signal-containing eigenspace of the hidden feature covariance.

**Strengths:**

The authors connect popular tools in interpretability research to matrix estimation results and spectral theory. The main idea may be relevant to efforts to improve interpretability tooling, and may guide the design of new probes. The authors avoid overly-technical formalism, although this sometimes impedes clarity. The authors perform empirical studies in synthetic settings to illustrate their argument.

**Weaknesses:**

1) The results do not deliver on the promise of proposing a practical diagnostic. The diagnostic requires omniscient knowledge (or at least an estimate) of the Fisher estimation error, but the authors do not provide an algorithm to obtain it. This shortcoming is highlighted in the experiments, which are restricted to synthetic settings in which the Fisher estimation error is known a priori.
2) The main result appears to be Theorem 4.1. However, this theorem appears to be a compilation of well-known results in matrix approximation theory. Concretely, it is an example of the spectral condition for estimating the top-k subspace of an unknown (uncentered) covariance matrix from samples.
3) The introduction makes several strong guarantees without stating the assumptions required. For example line 35 "ensures early interpretability," line 36 "guarantees clear separation," line 45 "prevents entanglement and instability," line 49, 50, 52 "ensures... guarantees...." Many of these guarantees use terms that are not defined in the introduction (or seemingly anywhere else in the paper), e.g., "robustness", "stability", "entanglement."
4) The claim in line 145 is unsupported. In fact, it is not clear how the label enters at all; indeed, the label $Y$ is not mentioned anywhere after being defined. Line 214 makes a similarly confusing claim. Line 146 makes a different confusing claim -- centering the covariance should have a very small semantic effect (since centering is a rank-1 perturbation).
5) There are a few sections with typos, incomplete sentences, misplaced sentences, etc. For example, line 68, line 300. Lines 40-53 are quite repetitive (is this intentional?)
6) The connection between the theory and practically-relevant settings is tenuous at best. At the least, it would be useful to see an empirical check of the SIP assumptions in natural settings.

**Questions:**

1) The terms "robustness" and "stability" are used throughout, but never defined. What perturbation is the probe robust to?
2) Line 267 posits a "clear separation between important features in the data." Is this a modeling assumption? I'd think it's not satisfied in practice. Is this related to the "discriminative subspace" mentioned in line 16?
3) Is there an assumption that the top subspace of h(X) is strongly correlated with Y(X)? I don't believe this is stated anywhere.

---

### Official Review · Reviewer_ik22 · 2025-10-31

**Soundness:** 2
**Presentation:** 2
**Contribution:** 2
**Rating:** 2
**Confidence:** 2

**Summary:**

The paper proposes a Spectral Identifiability Principle (SIP) for judging when linear probes on fixed representations will be reliable. Let $\Gamma=\mathbb{E}[hh^\top]$ be the (uncentered) second-moment operator and $\Delta$ the estimation noise of $\widehat{\Gamma}$. SIP posits that if there is a clear eigengap, then probe performance should be stable whenever the inline condition $\Delta < \operatorname{gap}(\Gamma)$ holds (with stability argued via a Davis–Kahan–style perturbation argument). Empirically, the paper only demonstrates synthetic Gaussian/Student-$t$ mixtures (varying eigengaps, sample size, and clipping) and does not validate on modern SSL/LLM features or compare against established, task-agnostic spectral diagnostics—α-ReQ (eigenspectrum-decay coefficient) and RankMe (effective rank)—that already correlate with linear-readout performance. This leaves the contribution primarily theoretical/expository rather than a validated practical diagnostic.

**Strengths:**

1. **Clear spectral lens:** Concise reduction of probe stability to a checkable inequality, stated inline as $\Delta < \operatorname{gap}(\Gamma)$, and a familiar Davis–Kahan control $|\sin\Theta(\widehat U, U)|_2 \le |\widehat\Gamma - \Gamma|_2/\operatorname{gap}(\Gamma)$.

2. **Heavy-tail discussion:** The “winsorized/clipped” extension and sample-complexity narrative acknowledge realistic variance issues; even though confined to mixtures, this section clarifies where Davis–Kahan degrades.

3. **Deterministic→probabilistic bridge:** The split-sample proxy for $\Delta$ and randomized SVD form a reasonable recipe if validated on real features.

**Weaknesses:**

1. **Experimental scope is not commensurate with the claims.** All results are on synthetic distributions. There are no evaluations on modern representations (e.g., SimCLR/MoCo/Barlow Twins/MAE, ViT, LLM layers) and no downstream linear-probe tasks (ImageNet-linear, CIFAR-10/100, VTAB) that would establish SIP’s practical value.

2. **Missing baselines from the representation-quality literature.** The paper does not compare against—or even properly cite—

     α-ReQ: eigenspectrum-decay coefficient α as a task-agnostic quality score predictive of linear downstream performance.

     RankMe: effective rank / spectral entropy that correlates with downstream JE-SSL performance and detects collapse.

     These are directly relevant spectral diagnostics; ignoring them undermines novelty and significance.

3. **Label alignment vs. unsupervised spectra.** SIP’s $\Gamma=\mathbb{E}[hh^\top]$ is label-free, yet the paper claims to identify a “discriminative subspace.” On real models, top-variance directions are not guaranteed to align with label-relevant components; α-ReQ and RankMe explicitly study when spectral structure predicts linear readout, with caveats.

4. **Terminology and scope.** Referring to $\Gamma$ as “Fisher” is nonstandard; several statements about “guarantees before training” exceed what $\Delta/\operatorname{gap}(\Gamma)$ can justify without continual re-estimation.

**Questions:**

1. On frozen SSL backbones (ResNet-50 / ViT-B with DINOv2, Barlow Twins, etc.), does the inline ratio $\Delta/\operatorname{gap}(\Gamma)$ correlate with ImageNet-linear accuracy? Please report calibration plots and compare to $\alpha$-ReQ and RankMe on the same features.

2. Under what conditions does the top-$k$ eigenspace of $\Gamma$ align with label-relevant directions? If misaligned, would a between-class operator make the $\Delta < \operatorname{gap}(\cdot)$ rule better grounded?

3. Please sweep $k$, clipping quantiles, and show how the empirical excess-error proxy tracks $(\Delta/\operatorname{gap}(\Gamma))^{\kappa}$ on real features, not only synthetic data.

4. Can the method complement $\alpha$-ReQ (power-law decay) or RankMe (effective rank) by flagging regimes where those metrics succeed but $\Delta/\operatorname{gap}(\Gamma)$ fails (or vice-versa)?

---

### Comment · Area_Chair_hcAK · 2025-11-26

Dear authors, we note that no author response has been posted during the author response window (Discussion Period: Nov 12 – Dec 3, 2025). To ensure a fair review, please post a reply addressing the reviewers' main concerns on this forum by Dec 3, 2025.

---

> ### Author Response · Authors · 2025-11-26
> **Response to Reviews**
>
> We sincerely thank the reviewers for the detailed comments.
> We would like to clarify one point regarding the technical content: all mathematical derivations, proofs, and spectral conditions in the submission were developed and checked by the authors themselves. Any language polishing tools were used only for grammar/wording, and we take full responsibility for all formulations. The theoretical results and experimental code are entirely our own work.
>
> Regarding the assumptions: they were intentionally stated in full generality to make explicit the standard conditions needed for subspace estimation (concentration of the empirical operator, eigengap, margin). While Theorem 4.1 invokes a subset of them directly, the remaining ones support the derivation of the excess-risk connection and the distributional assumptions for the synthetic settings. We agree that this could have been stated more clearly, and we will consolidate the assumptions to avoid the impression of inconsistency.
>
> On the choice of GLM-style synthetic models: this was deliberate, not because they are the only relevant setting, but because they allow us to compute the true Fisher operator exactly. This enables a precise evaluation of ∆ and the eigengap, which is impossible for modern neural models. We fully agree that experiments on real SSL/ViT/LLM features would strengthen the empirical side and will include such comparisons in a revision.
>
> We appreciate the reviewers’ feedback and will revise the narrative to more accurately match the scope of the theorem, emphasizing its role as a sufficient spectral condition rather than a general interpretability diagnostic.

---

### Meta-Review · Area_Chair_s9Us · 2025-12-17

**Summary:**

The reviewers unanimously rejected the paper due to a severe lack of empirical validation and theoretical novelty. They noted that the proposed Spectral Identifiability Principle (SIP) was tested only on synthetic Gaussian mixtures, failing to evaluate modern neural network representations or compare against established spectral baselines like Alpha-ReQ and RankMe. Consequently, the strong claims regarding the method's ability to guarantee interpretability and stability during training were considered entirely unsupported, as no actual model training experiments were conducted.

Furthermore, the theoretical foundations were criticized as being merely a restatement of well-known matrix approximation results, specifically the Davis-Kahan theorem, without providing a practical diagnostic algorithm. Reviewers pointed out numerous logical inconsistencies, undefined terminology, and a lack of clarity regarding how label-free spectral analysis could identify label-relevant subspaces. One reviewer strongly suspected the manuscript was generated by an LLM, citing erratic shifts in focus, leftover instructional text in the appendix, and significant formatting artifacts. The original AC agreed with this assessment.

**Reviewer Concerns:**

None of the concerns were addressed as the authors did not provide rebuttals - they only provided a fairly general defence of the paper in a short comment replying to the AC (which did insist that their formal analyses were not performed by an LLM).

**Reviewer Scores:**

The original scores were 2,2,2,0, and I don't think these would have changed.

---

### Decision · Program_Chairs · 2026-01-26

Reject